

# Identification of a pathogen causing fruiting body rot of *Sanghuangporus vaninii*

Weidong Yuan[1,*], Lin Ma[2], Xingkun Chen[3], Jiling Song[1] and Qing Chen[4,*]

[1] Hangzhou Academy of Agricultural Sciences, Hangzhou, Zhejiang, China
[2] Jiangsu Key Laboratory for Horticultural Crop Genetic Improvement, Institute of Vegetable Crops, Jiangsu Academy of Agricultural Sciences, Nanjing, Jiangsu, China
[3] College of Agronomy, Shandong Agricultural University, Taian, Shandong, China
[4] Zhejiang Agricultural Technology Extension Center, Hangzhou, Zhejiang, China
* These authors contributed equally to this work.

## ABSTRACT

*Sanghuangporus vaninii* is a medicinal macrofungus that is increasingly cultivated in China. During cultivation, it was found that the fruiting body of *S. vaninii* was susceptible to pathogenic fungi, resulting in significant economic losses to the industry. The symptoms of the disease occur in the initial stage of fruiting body development. The isolate YZB-1 was obtained from the junction of the diseased and healthy areas of the fruiting body. In order to verify the pathogenicity of YZB-1, its purified spore suspension was inoculated into the exposed area nearby the developing fruiting body of *S. vaninii*. After 10 days, the same disease symptoms appeared in the inoculated area. Morphological identification and molecular analysis of rDNA ITS region confirmed that the isolate YZB-1 was identified as *Trichoderma virens*. The temperature stability assay revealed that the mycelia of YZB-1 grew the fastest at 25 °C, with growth slowing down gradually as the temperature increased or decreased. Dual-culture tests of *T. virens* and *S. vaninii* showed that the inhibition rate of *T. virens* on *S. vaninii* mycelium was the highest (79.01 ± 2.79%) at 25 °C, and more green spores were produced at the intersection of *T. virens* and *S. vaninii*.

## INTRODUCTION

*Sanghuangporus vaninii* (Ljub.) *Zhu et al. (2019)* and *Wu & Dai (2020)* is a species of Basidiomycota, Hymenochaetales, Hymenochaetacae, *Sanghuangporus*, of which fruiting body is commonly known as "Sanghuang" in China. Sanghuang has been recorded in historical studies such as "On Medicinal Properties" and "Compendium of Materia Medica" (*Kim et al., 2004*; *Sun et al., 2006*; *Song et al., 2019*). *Sanghuangporus vaninii* is considered as one of the medicinal macrofungi due to its excellent efficiency in treating dysentery and blood insidiousness, anti-tumor, hypoglycemic, anti-oxidative, and immune-enhancing effects (*Song et al., 2020*). It has been a hot topic in the research and development of pharmaceutical preparations and health products industries in China and some other countries (*Che et al., 2005*; *Gao, Zhang & Yu, 2014*). The development of

Corresponding author
Lin Ma, 20090029@jaas.ac.cn

Sanghuang industry promotes the revitalization of rural economy in China. In 2021, the production of Sanghuang increased to 300 t, and the industry was attached great importance by the government (*Yang et al., 2023a*).

In China, Sanghuang and other mushrooms are grown using facilities cultivation techniques. Once the facilities are built, the same variety of mushroom is cultivated every year. Some even achieve annual cultivation in facilities by controlling temperature or rotating mushrooms suitable for different seasons, to improve facility utilization and obtain higher economic benefits (*Yang et al., 2023b*). However, as the cultivation years increase, the occurrence of diseases has a great impact on mushroom cultivation, reducing the quality and yield. A large number of diseases have been reported in mushroom cultivation, such as wet bubble disease caused by *Mycogone perniciosa* in white button mushrooms (*Agaricus bisporus*) (*McGee, 2018*; *Yang et al., 2021*), dry bubble disease caused by *Verticillium fungicola* in white button mushrooms and oyster mushroom (*Murmu, Maurya & John, 2020*), cobweb disease caused by *Cladosporium* spp. in oyster mushrooms (*Oyebamiji et al., 2018*; *Gea, Navarro & Suz, 2019*), and white mold disease caused by *Paecilomyces penicillatus* in morels (*Yu et al., 2022*). In addition to fungal pathogens, *Pseudomonas tolaasii* is consistently associated with mushroom brown blotch disease (*Ghasemi et al., 2021*), while *Ewingella americana* has been reported as a pathogenic bacterium of brown rot disease on shiitake mushroom (*Na, Luo & Yu, 2021*). However, despite the history of more than 2000 years of Sanghuang in China, diseases occurring during the process of *S. vaninii* cultivation have not been reported so far due to its short time of artificial cultivation.

In recent years, artificial cultivation of *S. vaninii* has made great progress and the cultivation scale is expanding (*Yang et al., 2023a*). However, the disease problem is becoming more prominent. From 2018 to 2021, we investigated cultivation companies where the disease occurred and found that the incidence of fungal disease in the cultivation bags of *S. vaninii* was as high as 30–70% in Hangzhou city, Zhejiang province of China. The symptoms of these diseases are basically the same, occurring in the initial or developing stage of *S. vaninii* fruiting bodies, preventing fruiting body formation, or causing brown to dark brown lesions on the fruiting body. The occurrence of this disease influences the quality and yield of Sanghuang, causing great economic losses to producers and becoming an important restriction factor of the Sanghuang industry.

In this study, we observed and described the symptoms of diseases in *S. vaninii* cultivation bags, isolated and identified pathogens using morphological characteristics and phylogenetic analysis with a combination of rDNA ITS genetic regions. The temperature stability of the pathogen was analyzed by *in vitro* test.

## MATERIALS AND METHODS

### Isolation and purification of pathogens

Disease symptoms of *S. vaninii* were observed in a greenhouse at Hangzhou Academy of Agricultural Sciences, located in Zhejiang province, China (120°0′88″E, 30°1′63″N) between late June and late July 2020. Ten diseased cultivation bags were collected, and samples were taken from the junction of the diseased and healthy areas of each bag and

plated onto potato dextrose agar (PDA) containing 0.25 g chloramphenicol. The plates were then incubated at 25 °C. After 7 days of incubation, agar blocks (5 mm in diameter) were cut from the growing edge of colonies and inoculated onto fresh PDA, and this process was repeated several times to obtain putative pure pathogens.

## Pathogenicity assay

To conduct the pathogenicity assay, we prepared a conidial suspension ($1 \times 10^6$ spores/mL) using five representative isolates. At the end of the vegetative growth stage of *S. vaninii*, a semicircle was cut in the middle of the plastic bags to somatic part of the mycelia in the air. Then, 500 µL of the pathogen's conidial suspension was inoculated into the areas surrounding the initial fruiting bodies of *S. vaninii*. The bags were incubated for 10 days at 25 °C and a relative humidity of 98%, and each isolate was tested in triplicate. Uninoculated bags were used as controls. Disease symptoms were observed and recorded, and the pathogens were isolated again from the diseased sites to confirm their morphological characteristics.

## Morphological identification

To identify the fungal pathogens, ten representative isolates were cultured on potato dextrose agar (PDA), CMD (cornmeal agar 20 g, dextrose 20 g, agar 20 g with 1 L distilled water) and SNA ($KH_2PO_4$ 1 g, $KNO_3$ 1 g, $MgSO_4 \cdot 7H_2O$ 0.5 g, KCl 0.5 g, glucose 0.2 g, sucrose 0.2 g, agar 15 g with 1 L distilled water) (*Jaklitsch, 2009*), and incubated at 23 °C under a 12-h light/dark cycle. The structure of conidiophores, phialides, and conidia were observed and measured using a Zeiss Axiophot 2 microscope equipped with an Axiocam CCD camera and Axiovision digital imaging software (Axio-Vision Software Release 3.1., v.3–2002; Carl Zeiss Vision Imaging Systems, Jena, Germany), as previously described (*Tomah et al., 2020*).

## Molecular analysis

To analyze the ITS region and the genes involved in taxonomy, ten isolates of pathogens were grown in 100 mL potato dextrose broth (PDB) on a shaker at 180 rpm, 25 ± 1 °C for 3 days. Genomic DNA was extracted using the Ezup Column Bacteria Genomic DNA Purification Kit (Sangon Biotech Co., Shanghai, China) according to the manufacturer's instructions. The ITS rDNA regions were amplified using the primer pairs ITS5 (5′ GGAAG TAAAAGTCGTAACAAGG3′) and ITS4 (5'TCCTCCGCTTATTGATATGC3′) (*Jiang et al., 2016*). The purified PCR product was sequenced in both directions and edited by BioEdit 7.1.3.0. and compared with homologous sequences available in the NCBI databases using BLAST.

Multiple alignment of the ITS rDNA sequences of this study and sequences from NCBI database (type strains of *Trichoderma* species, containing some species reported to be harmful to edible mushroom and some species closely related to the isolated strains) was carried out using Clustal W and a phylogenetic tree was constructed using MEGA 6. The evolutionary history was inferred by using the maximum likelihood (ML) method based on the Jukes-Cantor model (*da Silva et al., 2017*). The ML method was used to

construct the phylogenetic tree with 1,000 bootstrap frequency. The type strain *Sphaerostilbella lutea* CBS 405.59 was used as the outgroup (*Perera et al., 2023*).

### Temperature stability assay

Temperature stability was assessed by investigating *in vitro* mycelial growth at different temperatures. Isolate disks (5 mm diameter) were cultured on PDA plates and incubated in the dark at 5 °C, 15 °C, 25 °C, 30 °C, and 35 °C, each temperature treatment three replicates respectively. After 48 h, the diameters of the mycelial colonies were measured. Through diameter comparison, the temperature range suitable for the growth of the isolate was selected to continue the next experiment.

The inhibition of pathogenic isolate on mycelial growth of *S. vaninii* at different temperatures was observed by dual-culture test (*Zang et al., 2023*). Disks (5 mm diameter) of *S. vaninii* were placed on one side of PDA plates and incubated in the dark at 15 °C, 25 °C, and 30 °C, each temperature treatment three replicates respectively. Seven days later (to compensate for the slower growth of *S. vaninii*), disks of pathogenic isolate were placed on the other side and continued to incubate at the same temperature. The plates with only one disk of *S. vaninii* without pathogenic isolate were used as controls. After another 9 days, the radius of the mycelial colonies of *S. vaninii* was measured.

Analysis of variance (ANOVA) was done using SPSS 20.0 software program (SPSS Inc., Chicago, IL, USA). Mean value and standard deviation of each experiment were grouped according to S-N-K multiple range test with significance level of 5%. Dunnett's test ($P < 0.05$) was also used to compare treatment plots with positive and negative control plots in the experiments.

## RESULTS

### Disease symptoms and pathogen isolation

During the process of artificial cultivation, disease symptoms typically occurred around the timing of fruiting body production of *S. vaninii*. After the somatic growth of *S. vaninii* in a cultivation bag ended, a semi-circular area in the middle of the bag was cut to expose a part of mycelia for the development of fruiting bodies. Pathogen contamination manifested as white hyphae covering the surface of the exposed area or by infecting the initial small fruiting body. Subsequently, green spores appeared on the white mycelium (Fig. 1A). The entire exposed substrate or the fruiting body could be covered by the pathogen mycelium (Fig. 1B), thus preventing development or further development of the fruiting body. The disease symptoms were similar to those caused by *Tricoderma* spp. in green mold disease on other mushrooms. After purification, five representative single-spore isolates (YZB-1 to YZB-5) were collected for pathogenicity testing and identification.

### Pathogenicity tests

A spore suspension of the five isolates was inoculated into the exposed area nearby the developing fruiting body of *S. vaninii*, and white hyphae developed rapidly. Ten days after inoculation, a lot of hyphae with a green mold layer covered the exposed substrate and surrounded the developing fruiting body (Fig. 1C). All of the inoculated bags showed the

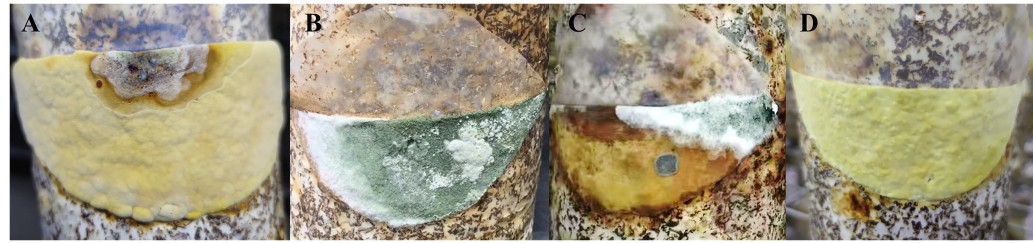

**Figure 1 Disease symptoms during the cultivation of *S. vaninii* and after artificial inoculation.**
(A and B) During the cultivation of *S. vaninii*. Pathogen hyphae covering the surface of the initial fruiting body and exposed substrate. (C) After inoculation. Hyphae inoculated with isolate YZB-1 covering the surface of the substrate and surrounding fruiting body. (D) Normally growing *S. vaninii* fruiting body.                                   

same symptoms as the natural incidence, whereas the control treatment remained symptomless. The five isolates were separated from the inoculated bag again (YZB-1-P to YZB-5-P).

## Morphological identification of pathogens

The colony characteristics of all ten isolates were similar. On PDA, the colonies were floccose with massive conidiation covering the whole surface of the plate (Fig. 2A). On CMD, isolates had a flat colony with aerial mycelium (Fig. 2B). Conidiophores and conidia were produced concentrically or near the margin of the plate. On SNA, they were relatively sparse (Fig. 2C). Conidiophores were gliocladium-like, arising from aerial hyphae, straight, 42–75 µm long (*n* = 30), generally unbranched (Fig. 2D), and sterile near the base, branching irregularly near the tip, with each branch terminating in a whorl of 3–6 phialides; metulae and phialides arose at narrow angles. Phialides were lageniform or ampulliform, 8.5–9.0 × 3.9–4.2 µm at the widest point. Conidia were green, smooth, subglobose, 4.2–4.5 × 3.9–4.0 µm (Fig. 2E). The isolates were similar to *T. virens* Gli 21, as described by *Chaverri, Samuels & Stewart (2001)*. They are markedly different from the reported *Trichoderma* species in terms of spore size, color and location of colonization, phialides morphology and number of branches, and so on (*Tomah et al., 2020*; *An et al., 2022*).

## Molecular analysis

The DNA from ten isolates was amplified using the primer pairs ITS5/ITS4. Sequence alignment results showed that the ITS nucleotide identity of all isolates was 100%. One isolate, YZB-1, was selected for subsequent analysis, and the ITS fragments were approximately 630 bp in length. The accession number in GenBank is MZ220425.1. Phylogenetic analysis was performed using ITS sequences from 31 type strains of *Trichoderma* species and one outgroup type strain *Sphaerostilbella lutea*. The resulting phylogenetic tree showed that all strains were separated into different clades (Fig. 3), and most reference strains could be distinguished on the species level. Strain YZB-1 was clustered together with *T. virens*. These data confirmed that YZB-1 is a member of *T. virens*.

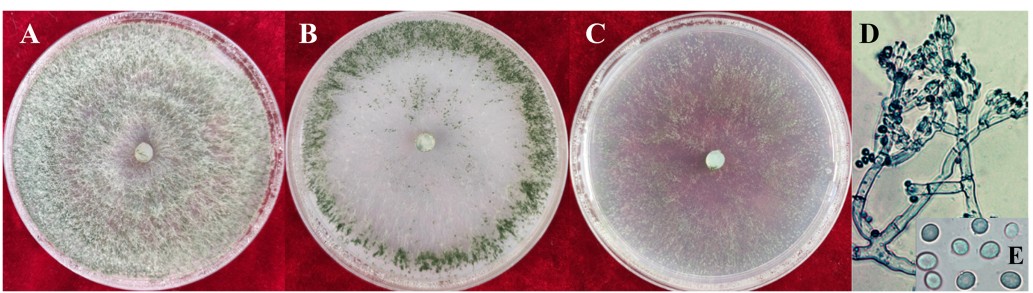

**Figure 2 Colonies and microscopic photographs of pathogenic fungi.** YZB-1 grown on PDA, CMD or SNA in 9-cm-diam Petri dishes under 12 h darkness/12 h light for 7 d. (A) On PDA. (B) On CMD. (C) On SNA. (D and E) Conidiophores and phialides conidia. D = 100 μm; E = 10 μm.

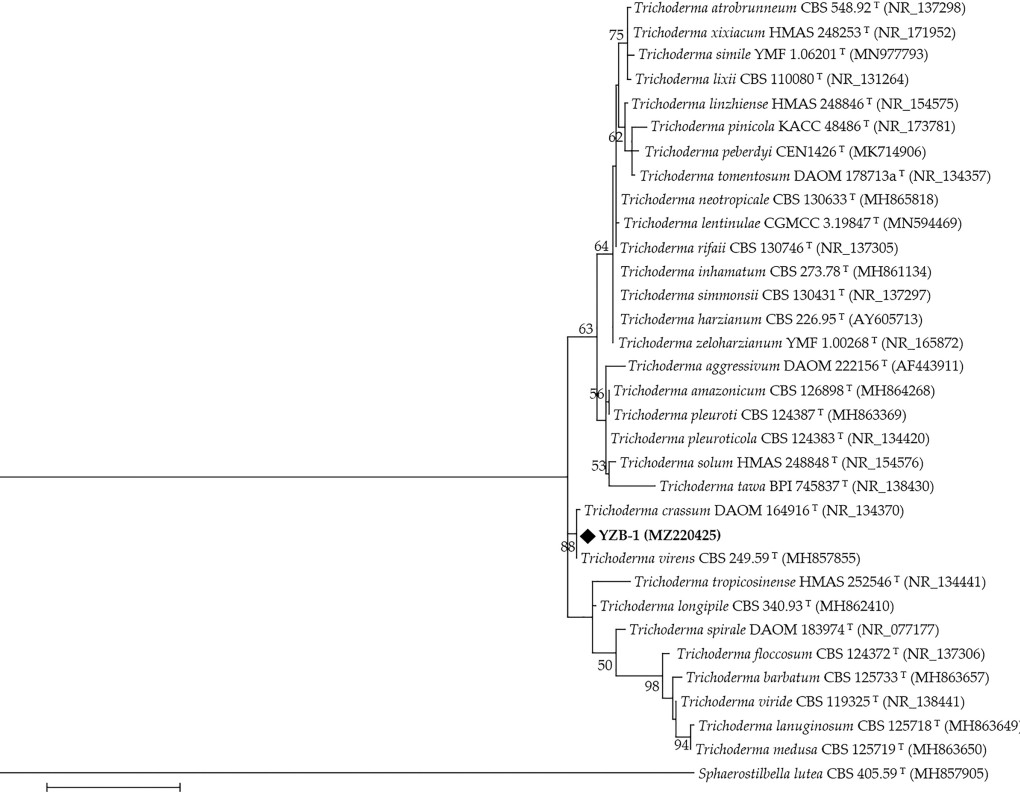

**Figure 3 The phylogenetic tree generated from the ITS sequences of *Trichoderma* spp.** Branch values lower than 50% were omitted.

## Temperature stability assay

The mycelial growth of *T. virens* strain YZB-1 was significantly affected by different incubation temperatures (Fig. 4). The mycelia grew fastest at 25 °C, with an average colony diameter of 57.67 ± 2.52 mm. At temperatures above or below 25 °C, mycelium growth gradually slowed down. At 5 °C, the mycelia stopped growing. Dual-cultures of *T. virens* and *S. vaninii* were performed at temperatures suitable for pathogen growth (15 °C, 25 °C, and 30 °C). The inhibition rate of *T. virens* on *S. vaninii* mycelium was highest when

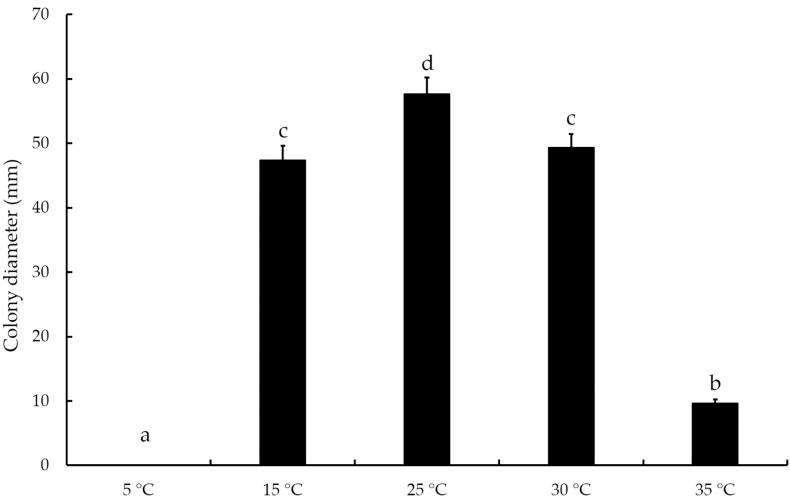

**Figure 4 The diameters of *T. virens* strain YZB-1 at different temperatures.** The error bars indicate the standard deviation, and different letters indicate significantly different values ($P < 0.05$).

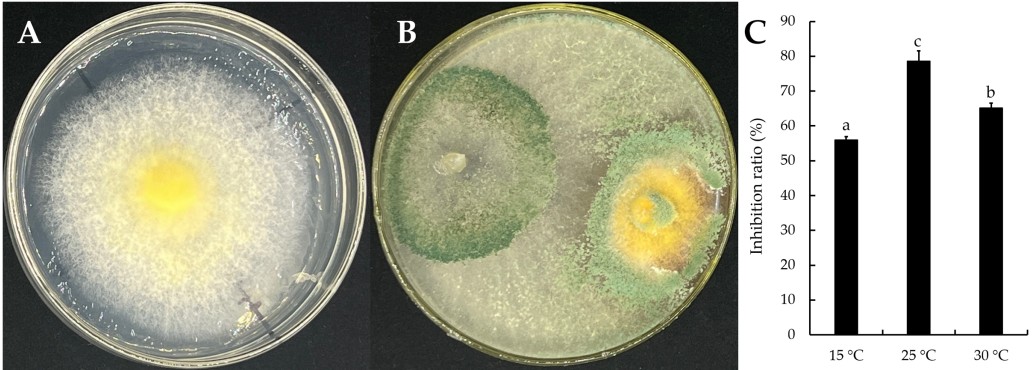

**Figure 5 The dual-culture of *T. virens* strain YZB-1 and *S. vaninii*.** (A and B) *S. vaninii* (A) and *T. virens* × *S. vaninii* (B) were incubated at 25 °C. (C) The inhibition ratios of *S. vaninii* by *T. virens* at different temperatures. The error bars indicate the standard deviation, and different letters indicate significantly different values ($P < 0.05$).

incubated at 25 °C (79.01 ± 2.79%), with significant differences in inhibition rates at the three temperatures (Fig. 5C). *Trichoderma virens* not only occupied the medium surface more quickly with mycelial growth but also produced more green spores at the intersection of *T. virens* and *S. vaninii* (Fig. 5B).

## DISCUSSION

*Sanghuangporus vaninii* is a renowned oriental medicinal mushroom, known in China as "Sanghuang," in Japan as "Meshimakobu," and in Korea as "Sangwhang" (*Chen et al., 2019*). Its fruiting body, also called yellow medicinal polyporus or basidiocarp, grows on the trunk of *Populus* sp. Linn., and is prized for its anti-tumor activity due to the bioactive protein-polysaccharide complex it contains (*Oh & Han, 1993*). However, Sanghuang occurs naturally in rare instances, making it highly valued. As a result, there has been

extensive research on the artificial cultivation of *S. vaninii* (*Wang et al., 2018*; *Hur, 2008*). To achieve the formation of fruiting bodies, indoor temperature ranging from 31–35 °C and over 96% relative humidity are ideal, conditions that are also suitable for the occurrence of diseases (*Hong, Sung & Nam, 2004*).

*Trichoderma* green mold in edible basidiomycetes has been well known for some time (*Hatvani et al., 2012*). Among the most significant diseases affecting the most commonly cultivated mushrooms worldwide, such as *P. ostreatus* and *A. bisporus*, are those caused by some *Trichoderma* species, including *T. guizhouense*, *T. harzianum*, *T. pleuroticola*, and *T. aggressivum* (*Bisset et al., 2015*; *Chaverri et al., 2015*; *Kosanovic, Grogan & Kavanagh, 2020*; *Turgay et al., 2023*). However, *T. virens* has been rarely reported to infect edible basidiomycetes. In this study, we found that *T. virens* colonized the mycelium of *S. vaninii*, with the infection being limited to the fruiting body stage. To our knowledge, this is the first report of green mold disease caused by *T. virens* in *S. vaninii* cultivation.

The antifungal mechanism of *Trichoderma* spp. against fungi has been reported because of their biocontrol functions. *Trichoderma* spp. control microorganisms through competition, parasitism, antibiotic action, synergistic antagonism, and other mechanisms (*Contreras-Cornejo et al., 2016*). Compared to pathogenic microorganisms, *Trichoderma* spp. have faster growth and reproduction rates, stronger decay ability, and wider adaptability. The optimal growth temperature for *Trichoderma* spp. for biocontrol is 25–30 °C (*Daryaei et al., 2016*). They achieve a fungistatic effect by competing for the living space and nutrient resources of pathogens (*Alwathnani et al., 2012*). When *T. harzianum* and *Fusarium solani* were co-cultured, *T. harzianum* parasitized *F. solani* from multiple contact points and led to its death (*Amira et al., 2017*). Additionally, the *Trichoderma* group can degrade the cell wall of pathogens and absorb their nutrients by secreting a series of hydrolases, such as cellulase, glucanase, chitinase, and protease (*Mukherjee et al., 2013*). *Trichoderma* is beneficial in plant cultivation, but harmful in edible mushroom cultivation (*Kredics et al., 2021*).

As macroscopic fungi, the growth of edible mushrooms is also inhibited by *Trichoderma* species as aforementioned antifungal mechanism (*Velázquez-Cedeño et al., 2007*; *Abubaker, Sjaarda & Castle, 2013*). The optimal growth environment for *Trichoderma* is consistent with the mycelia growth and fruiting body formation environment of most edible fungi, which leads to its infection and harm to edible fungi during the mycelium and fruiting body stages (*Kosanovic et al., 2020*; *Ponnusamy et al., 2022*). This was confirmed by the results of both fruiting body inoculation and hyphal dual-culture experiments in the present study. There are few reports on the pathogenic mechanism of *T. virens* infecting the fruiting body of edible mushrooms, which may be related to parasitism and antibiotic action. The control of *Trichoderma* mainly relies on environmental control methods for prevention. Some safe agents (*Innocenti et al., 2019*) or biocontrol microorganisms (*Ma et al., 2019*) can be used to control *Trichoderma* during the hypha growth stage. However, the agent may have the potential to cause phytotoxicity (*Kwon et al., 2021*) or residues (*Li et al., 2022*) during the fruiting body growth stage.

## CONCLUSIONS

This study has confirmed that the pathogen responsible for fruiting body rot in *S. vaninii* is the isolate YZB-1 through pathogenicity assays. Based on morphological identification and molecular analysis of the rDNA ITS region, the isolate YZB-1 was identified as *T. virens*. *Trichoderma virens* not only infects the fruiting body and causes abnormal growth but also inhibits hyphal growth. Further confirmation is required to determine whether its infection process and pathogenesis are consistent with the above mechanism. Finding safe and effective control methods for *Trichoderma* disease in *S. vaninii* is crucial for future studies.

### Funding

This research was funded by a grant from the National Natural Science Foundation of China (No. 31901933) and the Sannongliufang Provincial Project of Zhejiang (CTZB-F190625LWZ-SN). The funders had no role in study design, data collection and analysis, decision to publish, or preparation of the manuscript.

### Grant Disclosures

The following grant information was disclosed by the authors:
National Natural Science Foundation of China: 31901933.
Sannongliufang Provincial Project of Zhejiang: CTZB-F190625LWZ-SN.

### Competing Interests

The authors declare that they have no competing interests.

### Author Contributions

- Weidong Yuan conceived and designed the experiments, performed the experiments, prepared figures and/or tables, authored or reviewed drafts of the article, and approved the final draft.
- Lin Ma conceived and designed the experiments, authored or reviewed drafts of the article, and approved the final draft.
- Xingkun Chen performed the experiments, analyzed the data, authored or reviewed drafts of the article, and approved the final draft.
- Jiling Song performed the experiments, prepared figures and/or tables, and approved the final draft.
- Qing Chen performed the experiments, analyzed the data, prepared figures and/or tables, and approved the final draft.

### DNA Deposition

The following information was supplied regarding the deposition of DNA sequences:
The isolate YZB-1 sequence is available at GenBank: MZ220425.1.
## Data Availability

The raw measurements are available in the Supplemental Files.

## Supplemental Information

Supplemental information for this article can be found online at http://dx.doi.org/10.7717/peerj.15983#supplemental-information.

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
