# Peer review of "Identification of a pathogen causing fruiting body rot of Sanghuangporus vaninii"

_PeerJ, doi:10.7717/peerj.15983_

## Round 0.1 · original submission · Major Revisions

The two reviewers pointed out the main issue concerning molecular analysis involving identification and phylogeny. Therefore, I strongly suggest that the authors revise all these parts of the manuscript in detail.

Reviewer 1 ·

Basic reporting

As far as I could assess as a non-native speaker professional english is used. Manuscript could be improved with more citations in introduction as well as more recent references on critical parts of the text. The figure of phylogenetic should be improved in order to present bootstrap values and a white background. Scientific names lack proper standardization.

Experimental design

The entire method related to molecular identification should be rewritten, specially indicating the selection criteria for species comparison, and more accurate description of phylogenetic procedures, from alignment to phylogeny. Statistical analyses would strengthen findings on growth and inhibition.
Overall methods should be described in further detail.

Validity of the findings

The morphological identification should be deeper explored, including a comparison with relevant species.
Phylogenetic analysis should be redone.

Annotated reviews are not available for download in order to protect the identity of reviewers who chose to remain anonymous.

·

Basic reporting

Lines 20-22, 36, 59, 71-72, 95, 140, etc: Suggestion for changing the terms fruiting body, sporocarp, and basidiocarp and term standardization. I recommend you use sporoma or sporome when singular and sporomata or sporomes when plural.

Italicize the genera and species names. Most of them were highlighted in the MS, but look at all to fix.

Lines 35-36: please fix the higher ranks names of S. vaninii. See comments in the PDF.

Experimental design

Molecular analysis:
-The authors could include all generated DNA sequences.
-The ingroup selection is not clearly stated.
-The outgroup is far related to the ingroup and this could influence the analysis. In the below paper, you will find genera closer to Trichoderma and Sphaerostilbella specimens appear to be enough.
https://link.springer.com/article/10.1007/s13225-022-00512-1
-In the presented phylogeny, the Bootstrap values are not shown in the branches. The authors presented only ML analysis and it is ok, but ML-BS values are not shown.

Validity of the findings

no comment

Additional comments

Other minor issues and suggestions are included in the PDF notes and/or highlights.

---

## Round 0.2 · Minor Revisions

One of the reviewers is still asking for some improvements and detailed it in his/her review. Therefore, I consider that the authors must follow those recommendations.

Reviewer 1 ·

Basic reporting

The authors improved the text, reference list, methods description and figures according to suggestions from the previous review round. However, I only still recommend improvements in method description (inclusion criteria of species on phylogeny, alignment algorithm and evolutionary model selection) and phylogenetic analysis (e.g. using more robust methods such as ML and BI). Whith this modifications, the MS will likely be suitable for publication in PeerJ.

Experimental design

-

Validity of the findings

Good

·

Basic reporting

A few format and text issues remain. They are marked in the PDF.

Experimental design

no comment

Validity of the findings

no comment

Additional comments

no comment

---

## Round 0.3 · accepted · Accept

Dear Authors

The paper is now ready in the way of being published! Congratulations!!!